# Difference in the Effect of Applying *Bacillus* to Control Tomato Verticillium Wilt in Black and Red Soil

**DOI:** 10.3390/microorganisms12040797

**Published:** 2024-04-15

**Authors:** Zhenhua Guo, Ziyu Lu, Zhongwang Liu, Wei Zhou, Shuangyu Yang, Jiayan Lv, Mi Wei

**Affiliations:** School of Agriculture, Shenzhen Campus, Sun Yat-Sen University, Shenzhen 518107, China; 15546100532@163.com (Z.G.); luzy5@mail2.sysu.edu.cn (Z.L.); wangzhongliu668@163.com (Z.L.); zhouw79@mail2.sysu.edu.cn (W.Z.); ysy251147@163.com (S.Y.); lvjy28@mail2.sysu.edu.cn (J.L.)

**Keywords:** biological control, *Bacillus*, soil type, tomato Verticillium wilt, soil microbial community

## Abstract

In practical applications, the effectiveness of biological control agents such as *Bacillus* is often unstable due to different soil environments. Herein, we aimed to explore the control effect and intrinsic mechanism of *Bacillus* in black soil and red soil in combination with tomato Verticillium wilt. *Bacillus* application effectively controlled the occurrence of Verticillium wilt in red soil, reducing the incidence by 19.83%, but played a limited role in black soil. *Bacillus* colonized red soil more efficiently. The *Verticillium* pathogen decreased by 71.13% and 76.09% after the application of *Bacillus* combinations in the rhizosphere and bulk of the red soil, respectively, while there was no significant difference in the black soil. Additionally, *Bacillus* application to red soil significantly promoted phosphorus absorption. Furthermore, it significantly altered the bacterial community in red soil and enriched genes related to pathogen antagonism and phosphorus activation, which jointly participated in soil nutrient activation and disease prevention, promoting tomato plant growth in red soil. This study revealed that the shaping of the bacterial community by native soil may be the key factor affecting the colonization and function of exogenous *Bacillus*.

## 1. Introduction

*Bacillus* species are among the most widely used microbial inoculants (MIs) and inhibit disease by antagonizing or competing with plant pathogens [1]. The inhibitory effect of *Bacillus* on the growth of pathogens involves various mechanisms, such as competing with pathogens for nutrients and ecological niches, producing antibiotics and hydrolases, producing siderophores, and inducing systemic resistance. In addition, *Bacillus* can also be used as a biofertilizer, which plays a role in nitrogen fixation, phosphorus solubilization, the use of potassium solution in the soil, and phytohormone provision [2]. The beneficial effects of *Bacillus* on plant growth and yield have been demonstrated in a wide variety of crops, including wheat, corn, soybean, sunflower, French bean, tomato, pepper, potato, cucumber, and many other crops [3]. However, the actual effect of field application of *Bacillus* on plant yield is often unstable and varies widely. This is because there are many abiotic and biotic factors that affect the colonization and function of *Bacillus* in natural environments [4].

The type of soil is an important factor in determining the structure of the microbial community because soil is the habitat of microorganisms, and the element content, texture structure, pH, and organic matter content affect the soil microbial community [5]. Schlemper et al. studied the effects of plant genotype, growth period, and soil type on the rhizosphere bacterial community composition of broomcorn and reported that soil type was the main factor affecting the bacterial community composition of sorghum, followed by plant growth period and genotype [5]. Wei et al. reported that in phosphorus barren paddy soil, the abundance of rare microbial groups harboring the alkaline phosphatase gene *phoD* increased, thus promoting the conversion of organic phosphates to inorganic phosphorus. However, in phosphate-fertilized paddy soil, microorganisms carrying *phoD* can convert phosphorus in the soil into microbial biomass P but not significantly promote soil organophosphorus mineralization [6].

The physical and chemical properties of soil and the microbial community shaped by soil strongly affect the survival and colonization of plants by exogenous inoculants, which is important for affecting the stability of exogenous functional bacterial agents for disease prevention and growth promotion. However, most of those previous studies focused on the effects of soil type on indigenous soil microbial communities and paid little attention to the effects of the soil environment and the underlying mechanisms on the effects of microbial agents, including *Bacillus*. In addition, the colonization process and regulatory mechanisms of rhizosphere probiotics are also not well understood. Therefore, studying the differences in application effects in different soil environments is highly important.

Tomato Verticillium wilt is a soil-borne disease caused by *Verticillum dahliae* and has recently become one of the major tomato diseases in the world, greatly reducing the quality and yield of tomatoes [7]. *Verticillium dahliae* infects the root tips and penetrates the vascular system of the host, gradually extending to the stem and other parts of the shoots, inducing characteristic symptoms such as foliar yellowing and wilting, vascular discoloration, growth reduction, yield losses, poor fruit quality, and plant mortality [8,9]. In addition, preventing and controlling fungal pathogens is difficult because they can cause dormant microsclerotia to form in the soil and can tolerate adverse soil conditions [10]. We constructed a *Bacillus* combination in the previous stage that has significant inhibitory activity against *Verticillium dahlia* [11], but the effect of controlling Verticillium wilt in practical application is different. Based on this, the *Bacillus* combination was applied in black soil and red soil to explore the difference in the control effect on controlling tomato Verticillium wilt and the growth of tomato, and we clarified the difference in colonization of *Bacillus* inoculants in the two soil environments. We hypothesized that (1) nutritionally, rich black soil has a more abundant and stable native microbial community than red soil; (2) *Bacillus* can colonize red soil effectively, thus effectively inhibiting the occurrence of tomato Verticillium wilt in red soil; and (3) the application of *Bacillus* can synergically inhibit Verticillium wilt by regulating the function of the tomato rhizosphere microbial community. These findings contribute to the scientific evaluation of the application potential of *Bacillus* agents in agricultural production and provide a new reference for the application of *Bacillus* agents in preventing and controlling diseases such as Verticillium wilt, increasing crop yield, and improving the soil ecological environment.

## 2. Materials and Methods

### 2.1. Experimental Materials

Two types of soil with high and low nutritional status were used in this study. The high-nutrient soil was a natural mountain black soil collected from Changbai Mountain, Jilin Province (126°26″ E, 41°56″ N). The low-nutrient soil was a natural mountain red soil collected from Fuyang city, Anhui Province (32°54″ E, 115°48″ N). All the soils were air-dried and passed through a 2 mm mesh before the pot experiments. The initial physical and chemical properties of the red soil and black soil are shown in Appendix A.

The tomato (Moneymaker) seeds were obtained from our own stocks. Tomato seeds were soaked and disinfected with 4% sodium hypochlorite for 15 min and then washed three times with water to remove residual disinfectant. Tomato seeds were uniformly placed on absorbent paper that filled the bottom of sterile plastic Petri dishes and was moistened by adding 5 mL of deionized water. The dish was placed in a 26 °C incubator for 4 days, after which the absorbent paper was changed once a day. Seeds of the same length were subsequently selected and transferred to a 72-well seedling tray containing nutritious soil for seedling cultivation. The plants were raised under 16 h light and 8 h darkness for 30 days and watered every 3 days.

### 2.2. Construction of the Bacillus Combinations

In addition to the microorganisms screened from tomato roots, this experiment also used the microorganisms with the function of promoting growth and preventing disease used in our previous research [11,12,13], all of which were stored in the laboratory of the College of Agriculture, Sun Yat-sen University. The strains tested were *Bacillus velezensis* D2, *B. velezensis* ZJ-11, *B. subtilis* A#, *B. subtilis* B#, and *Paenibacillus* 2#. Based on the principle of functional complementarity and stability of beneficial microorganisms, the ability to dissolve inorganic phosphorus solubilization, organic phosphorus solubilization, potassium solubilization, nitrogen fixation, IAA production, and biofilm formation were measured.

### 2.3. Assessment of Growth-Promoting Ability

Phosphate solubilization was determined by Bashan’s methodology [14]. Single colonies were picked and cultured overnight to obtain a fermented seed broth, which was then inoculated into phosphorus-free SRSM liquid medium supplemented with calcium phosphate medium at 1% inoculum. The strains were subsequently grown in phosphorus-free SRSM liquid medium supplemented with calcium phosphate. The amount of converted soluble phosphorus was determined by UV–Vis spectrometry to evaluate its ability to solubilize phosphorus.

Silicate solubilization was determined on the silicate bacteria medium by observing the oil droplet microspheres of the strain after culturing for 2–3 days at 30 °C [15]. Single colonies were removed from the LB plates and cultured in LB liquid medium overnight. After centrifugation at 6000 rpm for 1 min, the bacteria were removed from the supernatant and resuspended in normal saline, and the OD600 was adjusted to 0.5. Ten microliters of the bacterial suspension with an OD600 of 0.5 was plated onto silicate AGAR medium and grown for 2–3 days. The strength of the silicate solubilization ability of the strains was evaluated by the size of transparent droplets around the bacteria on the plate.

Nitrogen fixation ability was detected on nitrogen-free medium by observing the growth of the strains. Then, 100 μL of fresh bacterial suspension with an OD600 of 0.5 was coated on nitrogen-free solid medium and cultured at 37 °C for 2 days. The strength of the nitrogen-fixing ability was identified by observing the growth of the bacteria on the plate.

The IAA secretion capacity was measured by the Salkowski reaction [16]. Fresh seed solution from overnight culture was inoculated at 1% in LB liquid medium containing L-tryptophan (100 mg/L) and incubated for 24 h at 30 °C and 180 r/min with shaking. After centrifugation, the supernatant was mixed with Salkowski solution (50 mL of 35% HClO_4_ + 1 mL of 0.5 mol/L FeCl_3_) in the dark for 30 min, and the absorbance was measured at 530 nm. The absorbance value was substituted into the gradient concentration standard curve of indole acetic acid for calculation

The biofilm-forming ability of the beneficial combination was determined by crystal violet staining [17]. A total of 0.1 mL of OD600 = 0.5 bacterial suspension was inoculated into 12-well cell culture plates, 0.9 mL of TSB was added, and the cells were incubated for 24 h at 37 °C. After culture, the excess culture medium was removed, the plate wells were cleaned three times by adding 1 mL of sterilized PBS buffer to each well, and 500 µL of methanol was added to each well for fixation for 15 min. Then, the excess methanol in the culture wells was removed, and the plates were air-dried. Then, 0.5 mL of 1% crystal violet was added to each well to stain for 30 min, and the excess staining solution in the culture-empty well was removed. The excess staining solution was rinsed with deionized water, allowed to dry naturally, and eluted with 0.5 mL of 33% glacial acetic acid for 20 min. Finally, 0.5 mL of deionized water was added to each well for dilution, 200 µL of the eluate was aspirated into a 96-well plate, and the absorbance at 570 nm was measured using a microplate reader.

### 2.4. Growth Curve Determination and Inhibitory Effect Analysis

Ten microliters of each strain suspension was added to a sterile, capped 96-well cell culture plate containing 190 μL of LB medium. The microorganisms were cultured until the growth plateau phase at approximately 12 h. The OD600 in each well was measured with a SparkControl Magellan™ multimode microplate reader (TECAN, Männedorf, Switzerland) at 0.5 h intervals during this period. Eight replicates of the same strain were set up, and pure sterile LB medium was used as a negative control. For the *Bacillus* combination, growth curves within 12 h were determined by mixing four different bacterial solutions with an OD600 of 0.5 in equal proportions, with the same details as for the single-strain assay method.

The plate confrontation method was used to determine the inhibition rate of *Bacillus* and its combination against *V. dahliae*. Fresh *V. dahliae* agar was placed upside down in the center of the PDA plates and incubated at 25 °C for 5 days to allow smooth mycelial growth. On the fifth day, approximately 3 mm of sterilized circular filter paper was placed on 1/2 of the radius of the PDA plate, 20 μL of fresh *Bacillus* solution was added to the filter paper, and the culture was continued for 14 days. An additional control plate without *Bacillus* was set up and used to calculate the inhibition rate. The inhibition rate was calculated as the ratio of the difference between the control radius and the treated radius to the control radius.

### 2.5. Preparation of Bacillus Suspension and V. dahliae Spore Suspension

Single colonies were separated into four triangular bottles containing LB liquid medium and cultured in a constant temperature shaker at 30 °C at 180 r/min for 12 h, after which the fermentation liquid of the 4 strains was obtained. After centrifugation and washing the medium, bacterial suspensions were prepared, and the regulated cell concentration was 1.0 × 10^8^ CFU/mL. The combined *Bacillus* suspension was obtained by mixing the four bacterial suspensions at a 1:1:1:1 ratio by volume.

Four bacterial tablets (3 mm in diameter) were made by sterilizing holes punched from a fresh solid plate of *V. dahliae* on an ultraclean workbench, which were subsequently transferred to PDB media and cultured in the dark in a constant temperature shaker at 150 r/min at 25 °C for 6 days. The pathogen *V. dahliae* JR2 was obtained from Professor Tingli Liu of Nanjing Xiaozhuang University. The *V. dahliae* spore suspension was obtained by removing the broth and hyphae twice, after which the final concentration was adjusted to 1.0 × 10^7^ CFU/mL.

### 2.6. Pot Experiment

The tomato seedlings with consistent growth were transferred to 1300 mL plastic pots filled with 1000 mL of soil. The settings were as follows: *V. dahliae* was added to black soil only (B); *V. dahliae* and *Bacillus* were added to black soil (BT); deionized water was added to black soil for the control (BCK); *V. dahliae* was added to red soil (R); *V. dahliae* and *Bacillus* were added to red soil (RT); and *V. dahliae* water was added to red soil for the control (RCK). There were 10 potted plants in the BCK and RCK groups and 30 potted plants in the other treatments.

Tomato seedlings were planted in plastic pots after 30 days of growth. The *Bacillus* combination suspension was applied to the BT and RT treatments, the final density was adjusted to 5 × 10^6^ CFU/g, and other treatments were applied in sterile water. All the tomatoes were inoculated with a root-dipping *V. dahliae* spore suspension, except for those in the BCK and RCK treatments, after which the spore suspension was added to the soil of these treatments at a final density of 3 × 10^5^ CFU/g. After 5 days, the *Bacillus* combination suspension was added to the tomato roots at a density of 5 × 10^6^ CFU/g. After 30 days, the incidence of Verticillium wilt in tomato plants was statistically analyzed. The incidence was expressed as a percentage of the total number of plants with symptoms of Verticillium wilt. The aboveground height, dry weight, and fresh weight of the tomato plants were measured. The aboveground height was the vertical distance from the ground to the highest point of the natural plant.

### 2.7. Determination of Nutrients in Tomato and Soil

Tomato nutrients were measured following the Chinese standard determination (www.chinesestandard.net (accessed on 8 April 2024)): total nitrogen (NY/T 2419-2013) [18], total phosphorus (NY/T 2421-2013) [19], and total potassium (NY/T 2420-2013) [20] using an automatic Kjeldahl nitrogen analyzer K-375 (BUCHI, Flawil, Switzerland), a UV spectrophotometer HD-UV90 (HORDE, Weifang, China), and a flame spectrophotometer LBT-6400A (Shangfen, Shanghai, China), respectively.

Total nitrogen, total phosphorus, total potassium, organic matter, available phosphorus, and available potassium were determined in air-dried soil. Ammonium nitrogen and nitrate nitrogen were determined in fresh soil. The determination of the soil’s total nitrogen (TN) was carried out using the Kjeldahl method with the automatic Kjeldahl nitrogen analyzer K-375 [21]. The soil total phosphorus (TP) and total potassium (TK) were digested and extracted by HF-HCIO_4_ and determined by the molybdenum blue method (HJ 632-2011) [22] with a UV spectrophotometer (HD-UV90) and flame spectrophotometry (HJ 781-2016) [23] with a flame spectrophotometer (LBT-6400A), respectively. Soil available phosphorus (AP) was extracted by using 0.5 mol/L NaHCO_3_ and determined using the molybdenum blue method (NYT 1121.7-2014) [24] with an HD-UV90 ultraviolet spectrophotometer. Soil available potassium (AK) was extracted by using 1 mol/L CH_3_COONH_4_ and determined using flame photometry (NY/T 889-2004) [25] with a flame spectrophotometer (LBT-6400A). Nitrate (NO_3_^−^-N) and ammonium (NH_4_^+^-N) were extracted with 2 mol/L KCl and determined by the ultraviolet spectrophotometric method (GB/T 32737-2016) [26] with the ultraviolet spectrophotometer HD-UV90 the continuous plug-flow method (LY/T 1228-2015) [27] with the continuous flow analyzer AA3.

### 2.8. Determination of Soil Enzyme Activity

The activities of soil acid phosphatase and alkaline phosphatase were determined by the p-nitrophenyl phosphate method [28]. The activity of the soil protease was determined by the Folin method [29]. The activity of soil urease was determined by the Solarbio soil urease (S-UE) activity detection kit, and a unit of enzyme activity (U/g) was defined as the production of 1 µg of NH3-N per gram of soil per day. The soil catalase activity was determined by a Solarbio soil catalase (S-CAT) activity detection kit, and a unit of enzyme activity (U/g) was defined as the degradation of 1 mmol H_2_O_2_ per gram of soil per day. Soil sucrase activity was determined by a Solarbio soil sucrase (S-SC) activity detection kit, and a unit of enzyme activity (U/g) was defined as 1 mg of reducing sugar produced per gram of soil per day at 37 °C. The above soil enzyme activities were measured using a SparkControl Magellan™ multimode microplate reader.

### 2.9. Determination of Resistance-Related Enzymes in Tomato

The four treatments were reset: *Bacillus* was applied to black soil (BT0), deionized water was added to black soil as a control (BCK), *Bacillus* was applied to red soil (RT0), and deionized water was added to the red soil as a control (RCK). The final density of the *Bacillus* combination suspension in the soil was 5 × 10^6^ CFU/g when the tomato plants were transplanted. After 5 days, the tomato leaves were collected, frozen in liquid nitrogen, and stored at −80 °C. The determination of peroxidase, polyphenol magnesium oxide, superoxide dismutase, and phenylalanine ammoniase activity was performed with Solebol kits. There were three biological replicates for each enzyme assay.

### 2.10. Quantitative Analysis of Bacillus and V. dahliae

The absolute abundances of *Bacillus* and *V. dahliae* in the rhizosphere and bulk soil were determined via real-time quantitative PCR. The specific primers Bs16S1/Bs16R and ITS1-F/ST-Ve1-R [30] were used for the absolute quantification of *Bacillus* and *V. dahliae*, respectively. The plasmids containing specific genes of *Bacillus* and *V. dahliae* were diluted 10 times, after which the standard curve was obtained.

### 2.11. Microbial Community Analysis

Total soil DNA was extracted by using the MP FastDNA Spin Kit for Soil (MPBIO, Santa Ana, CA, USA), and the concentration and purity of the DNA were determined by using a NanoDrop One spectrophotometer (Thermo Scientific, Waltham, MA, USA). The specific primers 338F (5′-ACTCCTACGGGAGGCAGCAG-3′) and 806R (5′-GGACTACNNGGGTATCTAAT-3′) were used to amplify the variable region V3 + V4 [31]. The length and purity of the PCR products were determined by 2% agarose gel electrophoresis, and the targeted bands were excised with a universal DNA (TianGen, Beijing, China) purification recovery kit. The products were subsequently sent to Wekemo Tech Group Co., Ltd. (Shenzhen, China), for library construction and sequencing. The NEBNext^®^ Ultra DNA Library Prep Kit (NEB, Waltham, MA, USA) was used for library construction, and the Illumina MiSeq PE300 platform (Illumina, San Diego, CA, USA) was used for sequencing. All the raw sequences of all the samples were used to construct ASVs by controlling the quality, denoising, splicing, and removal of chimeras [32]. The ASV sequences were aligned with the SILVA database by the feature-classifier plugin of QIIME 2, and the confidence threshold was set to 0.7 to obtain the OTU table of the species. Mitochondrial and chloroplast sequences were removed using the feature-table plugin QIIME2.

The alpha diversity among the samples was characterized by the Chao1 richness index and the Shannon diversity index of each soil sample. The differences in beta diversity of the bacterial communities among the samples were characterized via principal coordinate analysis based on the UniFrac distance. The percentage of top-ranked phylum-level species in the different groups was presented as a grouped percentage pile-up map, and the grouped clustered heatmap was generated to show the variation in phylum-level species among the groups. Microbial network analysis was based on the random matrix theory (RMT) [33]. Correlations between species were calculated with the Pearson coefficient, the visualization of the microbial network was performed with Gephi 0.9.2, and the visualization of core species in the microbial network was carried out with Origin 2018 software. The difference at the genus level in each sample was analyzed by STAMP software (STAMP v2.1.3) [34]. The functional genes of the soil microorganisms were predicted by PICRUST2. The difference analysis of functional genes was performed by using STAMP. Variance partitioning analysis (VPA) and permutational multivariate analysis of variance (PERMANOVA) were applied to analyze the contributions of soil type, ecological niche, and change in the bacterial community caused by *Bacillus* in tomato roots. Correlations between environmental factors and microbial communities were evaluated by using Mantel and redundancy analysis [35]. The combined effect of soil physiochemical factors on *Bacillus* and *V. dahliae* was measured by multiple regression analysis on IBM SPSS Statistics 24 software. Linear regression was applied to analyze the association between the abundance of Verticillium and the Shannon index.

### 2.12. Statistical Analysis

After the confirmation of the assumptions of normality and homogeneity of variances, independent sample *t* tests were used to compare the significance level of the difference between the two datasets. The mean values plus standard errors and significance levels were calculated (* *p* ≤ 0.05, ** *p* ≤ 0.01, *** *p* ≤ 0.001). All analyses were conducted in SPSS 24.0 (SPSS, Inc., Chicago, IL, USA).

### 2.13. Data Availability

The raw sequencing data (the amplicon fastq files) are publicly available in the NCBI Sequence Read Archive (SRA) under the Bioproject number PRJNA1061767.

## 3. Results

### 3.1. Construction and Evaluation of Bacillus Combinations

Our previous study found that D2 and ZJ-11 strains had synergistic effects on promoting plant growth [12]. The growth promotion ability of the selected strains was analyzed, and the results showed that strains A#, B#, and 2# had a good ability to dissolve organic phosphorus, fix nitrogen, and produce IAA. *Bacillus amyloliquefaciens* 13-2 had an ideal ability to detoxify inorganic phosphorus, enable nitrogen fixation, and dissolve silicates (Appendix A). Based on the principle of functional complementarity, we constructed three multifunctional microbial combinations and evaluated the growth-promoting ability and biofilm formation ability, respectively. The results showed that the D2 + ZJ-11 + 13-2 + A# combination had the best growth-promoting ability and biofilm formation ability.

In addition, the four strains in the combination did not produce antagonistic effects when mixed in culture, and the inhibition rate of *V. dahliae* was the most significant, reaching 77.04% (Figure 1). Therefore, the *Bacillus* combination D2 + ZJ-11 + 13-2 + A# was used in the study of controlling tomato Verticillium wilt in black soil and red soil based on its multiple functions regarding disease prevention and growth promotion.

### 3.2. Effects of Bacillus Application on Tomato Biomass and Incidence of Verticillium Wilt

The effects of *Bacillus* application on the biomass and incidence of Verticillium wilt in different soils are shown in Figure 2. In black soil, the incidence of Verticillium wilt without *Bacillus* application was 53%, whereas that in the treatment group was 50%. These findings indicated that the application of *Bacillus* had no significant effect on the incidence of Verticillium wilt in black soil (*p* > 0.05). In red soil, the incidence of Verticillium wilt was 73.33% without *Bacillus* inoculation but decreased to 53.5% after *Bacillus* was applied, indicating that the incidence of wilt in red soil was significantly reduced by *Bacillus* application (* *p* ≤ 0.05).

Plants can resist invasion via resistance-related enzymes. The effects of *Bacillus* on inducing PPO, POD, PAL, and SOD in tomato plants were compared between black soil and red soil. After 5 days of *Bacillus* application, the PPO and POD activities were significantly greater in both the black soil and red soil (Appendix A). The activities of PPO and POD in black soil were 1.95 and 2.05 times greater than those in the control, while in red soil, they were 1.17 and 1.69 times greater than those in the control. However, inoculation with *Bacillus* did not significantly increase PAL or SOD activity in tomato plants. The exogenous application of *Bacillus* can enhance the defense response of tomato plants mainly by increasing the activity of PPO and POD.

In red soil, the plant height, fresh weight, and dry weight of tomato plants increased by 21.38% (* *p* ≤ 0.05), 46.83% (** *p* ≤ 0.01), and 84.62% (*** *p* ≤ 0.001), respectively. However, in black soil, the height and biomass did not significantly improve after *Bacillus* treatment (*p* > 0.05). Moreover, we observed that the biomass accumulation of tomatoes in black soil was greater than that in red soil, and the dry weights of the treated and control groups were 1.97 and 3.20 times greater than that of tomatoes in red soil, respectively (*** *p* ≤ 0.001). In conclusion, the results showed that the application of *Bacillus* could effectively inhibit the incidence of Verticillium wilt in red soil and promote the growth of tomato plants, while the effects of inhibiting Verticillium wilt and promoting the growth of tomato plants in high-nutrient black soil were not significant.

### 3.3. Effects of Bacillus Application on Tomato Nutrient Uptake in Two Soils

By analyzing the effects of *Bacillus* application on N, P, and K levels in soil and tomato plants, the activating effect of *Bacillus* on soil nutrients and the regulation of tomato nutrient absorption were clarified. As shown in Figure 2A, the total nitrogen content in the black soil was significantly greater than that in the red soil before planting. The soil total nitrogen content in both the black soil and red soil decreased significantly after tomato planting. After *Bacillus* was applied, the total nitrogen content and nitrate nitrogen content of the black soil significantly increased (*** *p* ≤ 0.001, * *p* ≤ 0.05; Figure 3A,E). However, there was no significant difference in the total nitrogen, ammonia nitrogen, or nitrate nitrogen content in the red soil (*p* > 0.05; Figure 3E–H). Moreover, we found that the average total nitrogen content of tomatoes grown in black soil was 1.74 times that of tomatoes grown in red soil, but the application of *Bacillus* had no significant effect on the total nitrogen content of tomatoes in either soil (*p* > 0.05; Figure 3I). These results indicated that *Bacillus* had no promoting effect on the absorption of nitrogen in tomato plants.

Similarly, the total phosphorus content in black soil was also greater than that in red soil, and the application of *Bacillus* had no significant effect on the total P content in soil (*p* > 0.05). However, the total P content in red soil decreased significantly after the tomatoes were planted (*** *p* ≤ 0.001), and the application of *Bacillus* significantly reduced the total P content in soil compared with that in the control treatment (* *p* ≤ 0.05). In addition, we found that the application of *Bacillus* did not significantly increase the total P content of tomato plants in black soil (*p* > 0.05) but did significantly increase the total P content of tomato plants in red soil by 50.14% (* *p* ≤ 0.05). The application of *Bacillus* had no significant effect on the activities of either acid phosphatase or alkaline phosphatase in black soil (Appendix A) and had no significant effect on the available phosphorus content; thus, it did not promote the absorption of available phosphorus by tomato plants (Figure 3B). However, the application of *Bacillus* significantly increased the alkaline phosphatase activity in red soil (Appendix A), and the total P content in red soil decreased significantly (Figure 3B). At this time, *Bacillus* transformed part of the insoluble phosphorus into available phosphorus, which was subsequently absorbed by the tomatoes, thus promoting an increase in the total P in the tomatoes (Figure 3J).

Additionally, *Bacillus* application had no significant effect on the total potassium concentration in black or red soil and had no significant effect on the total potassium concentration in tomato plants between the two groups (*p* > 0.05). In black soil, the soil organic matter content was significantly greater than that in the control group (*p* ≤ 0.05; Figure 3D), but there was no significant difference in the organic matter content in red soil (*p* > 0.05).

In general, the application of *Bacillus* strongly increased the content of nitrate nitrogen in black soil (*** *p* ≤ 0.001), significantly increased the content of available potassium (* *p* ≤ 0.05), and had no significant effect on the content of ammonium nitrogen or available phosphorus (*p* > 0.05) but did not promote the absorption of nitrogen, phosphorus, or potassium. In contrast, adding *Bacillus* to red soil promoted phosphorus absorption by tomato plants through the mineralization of phosphorus but had no significant effect on nitrogen or potassium absorption (*p* > 0.05).

### 3.4. Effect of Bacillus Application on the Bacterial Community Diversity and Abundance of V. dahliae

As shown in Figure 3A, the principal coordinates PCoA 1 and PCoA 2 were the components that caused the largest community differences, contributing 36.5% and 5.0%, respectively. The community structures of bacteria in black soil and red soil were significantly different (r = 0.71, *p* = 0.001). In addition, the alpha diversity of the bacterial community in tomato roots in black soil and red soil significantly differed (Appendix A), indicating that the microbial communities shaped by soils with different characteristics significantly differed.

A further analysis of the effect of *Bacillus* application on the tomato root bacterial community in both soils showed that the application of *Bacillus* had no significant effect on the bacterial community structure of the tomato rhizosphere or bulk soil in black soil (Figure 4B and Appendix A). However, in red soil, the application of *Bacillus* had significant effects on the bacterial community in both the rhizosphere and bulk soil. Figure 3C shows that the contribution of PCoA 1 was 36.5%, and that of PCoA 2 was 8.2%. The effect of *Bacillus* treatment on the bacterial communities in red soil was greater than that in black soil. This difference may be related to the significant colonization of *Bacillus* in red soil. The absolute abundance of *Bacillus* in the red soil in the treated group was significantly greater than that in the control group, while there was no significant difference in the black soil (Appendix A).

Moreover, the application of *Bacillus* also significantly inhibited the abundance of *V. dahliae* in the rhizosphere and bulk soils but had no effect on that in the black soil (Figure 4D,E). The correlation analysis of the Shannon index and the abundance of *V. dahliae* revealed that there was a significant negative correlation between them in both the rhizosphere and bulk soil. These findings suggested that *Bacillus* might affect the occurrence of Verticillium wilt in tomato plants by regulating the microbial community structure. Therefore, we further explored the complex interactions of the microbial communities through microbial network analysis and elucidated the mechanisms of soil nutrient changes and morbidity differences through functional prediction.

### 3.5. Effect of Bacillus Application on Microbial Networks

To explore whether *Bacillus* controls tomato Verticillium wilt by regulating the interaction of bacterial communities in the roots of tomato plants in the two soils, we constructed an empirical network and random network of bacteria based on the absolute abundance of the OTUs (Figure 5). The overall properties of the molecular ecological networks in the different groups are shown in Appendix A. By comparing the average clustering coefficient (avgCC), average path distance (GD), and modularity index, it was shown that the cohesion of empirical networks was significantly better than that of the random networks. These findings suggested that the node relationships identified in the empirical network were effective and reasonable. The total nodes, total links, avgK, connectedness, module, and avgCC in the black soil network were greater than those in the red soil network (Appendix A). In addition to the rhizosphere treatment group, the network density index in the black soil was greater than that in the red soil. This indicated that the scale of the bacterial community interaction network in black soil was greater than that in red soil, and the structure was more compact and complex.

Moreover, we found significant differences in the interaction networks between the treatment and control groups of the two soils (*p* < 0.001), which indicated that *Bacillus* application strongly disturbed the bacterial interaction network in both soils. The application of *Bacillus* could reduce the size of the bacterial interaction network in black soil, reduce the number of modules, and increase the degree of modularity. Moreover, avgCC increased and GD decreased in both the rhizosphere and bulk soils, indicating that there was a closer interaction network between the rhizosphere and bulk bacteria after *Bacillus* application. Conversely, the application of *Bacillus* to red soil increased the size of the bacterial network. Moreover, avgK, connectedness, density, and avgCC increased, while GD decreased. This suggested that *Bacillus* application resulted in tighter network interactions in the rhizosphere of red soil plants. However, in the bulk soil interval, avgK, connectedness, density, and avgCC increased and GD decreased, which indicated that the overall aggregation of the bulk soil network in red soil decreased because of the application of *Bacillus*.

### 3.6. Regulation of Bacterial Community Function by the Application of Bacillus

After the application of *Bacillus*, the number of genes in the bulk soil was greater than that in the rhizosphere, and the abundance of most genes increased, which was particularly obvious in the red soil, where the abundance of functional genes significantly increased in the rhizosphere and bulk regions, accounting for 3.6% and 13.9%, respectively (Figure 5).

In addition, the application of *Bacillus* significantly increased the abundance of dehydrogenase genes (Figure 6). In black rhizosphere soil, only dehydrogenase and phosphodiesterase genes [EC:3.1.4] were significantly enriched (Figure 6A), while phosphomonoesterase [EC:3.1.3] and protease-related genes were significantly enriched in bulk soil, but no significant changes in genes related to pathogen antagonism were found (Figure 6B). In the red rhizosphere soil, in addition to the significant enrichment of phosphomoesterase and dehydrogenase genes, the functional genes involved in pathogen antagonism were also significantly enriched (Figure 6C). Moreover, there was a greater number of functional genes significantly enriched in the bulk soil than in the rhizosphere soil. In addition, lysozyme and protease genes were significantly enriched (Figure 6D).

## 4. Discussion

We constructed a combination of multiple functional disease-preventing and growth-promoting *Bacillus* species and studied the ability of these bacteria to control tomato Verticillium wilt in black soil and red soil, hoping to determine the reasons for the differences in *Bacillus* spp. application in different soils. The results showed that, compared with black soil, red soil with the application of *Bacillus* could effectively inhibit the occurrence of Verticillium wilt and promote the growth of tomato plants. We found that this difference was closely related to the interaction between the microbial communities shaped by the two soils and *Bacillus*.

### 4.1. Application of Bacillus Can Significantly Inhibit the Occurrence of Tomato Verticillium Wilt in Red Soil

This study revealed that the incidence of Verticillium wilt in black soil was lower than that in red soil (Figure 2A). Previous studies have shown that the incidence of plant disease may be related to soil nutrients, and nutrient deficiency may not only lead to plant physiological diseases but also reduce pathogen tolerance [36], which may also explain why the incidence of Verticillium wilt in black soil is lower than that in red soil.

The two soils in our study showed large differences in N and P levels, and we are currently unable to judge which element is most favorable for the colonization of *Bacillus*. We believe that exogenous *Bacillus* can perform its growth-promoting function better under relatively low-nitrogen or -phosphorus conditions, which has also been confirmed in some studies [37,38,39]. In addition, soil pH is an important factor affecting the structure and function of soil microbial communities [40]. Studies have shown that neutral or nearly alkaline pH is beneficial for bacterial growth, while acidic pH is beneficial for fungal growth [41]. Therefore, regulating the optimal soil pH is one way to improve the therapeutic effect of beneficial soil bacteria on plant fungal diseases.

The effect of microbial agents on plant diseases is related to the colonization of beneficial microorganisms and the abundance of pathogens [42]. We used real-time fluorescent quantitative PCR to measure the absolute abundances of *Bacillus* and *V. dahliae* in both soils, and the results showed that the abundance of *Bacillus* in black soil was greater than that in red soil (Appendix A), while the abundance of *V. dahliae* was lower than that in red soil (Figure 4D,E). However, after the application of *Bacillus*, the abundance of *Bacillus* increased significantly only in the bulk region in black soil, while it increased significantly in both the rhizosphere and bulk regions in red soil (Appendix A), indicating that *Bacillus* colonizes red soil more efficiently than black soil. In addition, there was no significant decrease in the abundance of *V. dahliae* in black soil, while it decreased significantly in red soil (Figure 4D,E). Taken together, these findings indicated that inoculation with *Bacillus* significantly inhibited *V. dahliae* and subsequently prevented tomato Verticillium wilt. These results are consistent with studies showing that microbial agents can function only in restricted soils. For example, some studies have reported that bacterial inoculation has a much better stimulatory effect on plant growth in nutrient-deficient soil than in nutrient-rich soil [43]. Other results have also demonstrated the feasibility of inoculation technology using diazotrophic bacteria in micropropagated sugarcane varieties grown in soils with low to medium levels of fertility and infeasibility in soils with high levels of fertility [44]. In addition, increases in crop yield resulting from *Azospirillum* inoculation were consistently obtained when soil nutrients (N, P, K, and microelements) were limiting, while crop yields did not increase when nutrients were not limiting [45].

### 4.2. Application of Bacillus Significantly Promoted Phosphate Uptake in Red Soil

Plant rhizosphere growth-promoting bacteria, such as nitrogen-fixing bacteria and phosphorus- solubilizing bacteria, exist widely in soil and can activate soil nutrient elements and promote plant nutrient absorption [46]. This study revealed that the total nitrogen, nitrate, organic matter, and available potassium contents in black soil increased significantly after the application of *Bacillus*, but the nutrient content of the tomato plants did not significantly increase (Figure 3). These findings are similar to those of Dilfuza et al. who studied the influence of exogenous microbial agents on the growth of maize in soils containing different nutrients. Exogenous microbial agents promoted the absorption of nitrogen, phosphorus, and potassium in plants in nutrient-poor soil, while they activated only soil nutrients in high-nutrient soil and had no significant effect on the promotion of plant absorption. It is believed that the nutrient elements in the native soil met the requirements of plants [42].

The application of *Bacillus* in red soil significantly promoted the uptake of phosphate in tomato plants (Figure 3) because it improved the activity of phosphatase (Appendix A), which can dissolve fixed phosphorus into a form that can be directly absorbed by tomato plants, thus promoting the absorption of phosphorus and growth. Although the content of phosphorus in soil is high, most phosphorus is fixed, and very little phosphorus can be directly absorbed and utilized by plants [47]. Phosphate-solubilizing bacteria have been shown to promote plant growth and increase plant phosphorus content in low-phosphorus soils [48]. Wei et al. showed that when the soil phosphorus supply is sufficient, phosphorus is converted into microbial phosphorus, and in phosphorus-poor soil, microorganisms can convert organic phosphorus that is difficult for plants to absorb into available phosphorus [6]. This finding supported the finding that *Bacillus* application promoted phosphorus uptake only in red soil and not in black soil.

### 4.3. The Bacterial Community in Native Soil May Be the Key to Affect the Colonization of Bacillus and Then Affect the Disease-Preventing and Growth-Promoting Effect of Bacillus on Tomato

VPA was used to further analyze the effects of soil type, root niche (rhizosphere and bulk region), and *Bacillus* application on the tomato root bacterial community, and the results showed that soil type explained 39% of the community differences, which was greater than the effects of the root niche and *Bacillus* application (Appendix A). Moreover, the results of the soil bacterial alpha diversity and PCoA showed that there was a significant difference in the bacterial communities between the black soil and red soil (Appendix A). These results suggested that the different soil types in black soil and red soil had significantly different bacterial communities.

The efficient colonization of microbial agents is key to their function, but the colonization of microbial agents is subject to competition from indigenous microbial communities [49]. However, soil microbial communities are highly complex and diverse and are usually highly resistant to exogenous microbial invasion [50]. Our results showed that the ability of *Bacillus* to disturb the black soil bacterial community was weak but was greater in red soil. The literature reports that the colonization of soil by microbial agents is influenced by the diversity of microbial communities and the effectiveness of soil resources. As diversity increases, nutrients are being more effectively utilized, reducing the possibility of exogenous species occupying unoccupied ecological niches [51]. Therefore, the differences in the colonization and disturbance ability of *Bacillus* in black soil and red soil may be attributed to the greater diversity and richness of microbial communities in black soil. Moreover, the microbial co-occurrence network also reflects the response of native soil microbial communities to the application of *Bacillus*. We found that the scale and complexity of the bacterial network in black soil were greater than those in red soil, which could more effectively prevent the disturbance of *Bacillus*. This finding is consistent with the view that a complex soil microbial interaction network is more conducive to resisting the invasion of exogenous microorganisms [52].

Therefore, the positive effect of *Bacillus* on tomato growth in red soil was greater than that in black soil. Moreover, microbial agents may influence the interaction and stability of soil ecosystems by regulating the rhizosphere microbial community [53]. This study showed that the application of *Bacillus* increased the closeness of the rhizosphere bacterial communities in the two soils, especially by increasing the diversity and complexity of the interactions between bacteria in red soil and increasing the degree of aggregation. These findings are consistent with those of Liu et al. who similarly showed that the microbial agent B. subtilis Z-14 could enhance the network integrity of wheat rhizosphere bacterial communities [54]. However, due to the large number of microbial groups with large differences in abundance in the two soils, it is not yet possible to accurately determine the specific groups that can inhibit or promote the growth of Bacillus. Future studies may yield relatively reliable results based on a broader sample of soil areas.

The rhizospheric microbial community plays an important role in plant growth and is the first line of defense against disease infection [55]. The complex and compact microbial network helps soil microorganisms participate in nutrient cycling, promote plant growth, inhibit pathogens, and perform other functions [56]. Our results indicated that the abundance of functional genes involved in nutrient cycling changed more significantly in red soil than in red soil, which was consistent with the stronger disturbance of the bacterial community in red soil by *Bacillus*. Soil dehydrogenase can be used to evaluate the vitality of the soil microbial community and soil health [57]. After the application of *Bacillus*, the abundance of functional genes associated with dehydrogenase significantly increased in both the black soil and red soil, indicating that the exogenous application of *Bacillus* can improve the vitality of the soil bacterial community. In addition, the application of *Bacillus* increased the abundance of various nitrate reductase genes in the bulk soil within the black soil. Nitrate reductase can participate in the reduction in soil nitrite and provide a substrate for the conversion reaction to generate nitrogen (NH4+) [58]. The activity of this enzyme was also positively correlated with urease activity and the soil total nitrogen content [59,60], which supported the finding that *Bacillus* application improved soil total nitrogen and urease activity. In black soil, only a phosphomonoestase gene in the bulk soil was significantly enriched [EC: 3.1.3.73], while in the rhizosphere and bulk soil of red soil, not only was the phosphomonoestase gene enriched, but the functional gene of the alkaline phosphatase isozyme conversion protein [EC:3.4.11-] was also significantly increased. This explained the promotion of tomato phosphorus absorption by the application of *Bacillus* to red soil. In addition, after the application of *Bacillus*, the abundance of functional genes related to pathogen antagonism, such as streptomycin and lysozyme, and protease functional genes, such as subtilisin and trypsin, increased in the red soil bacterial community, which explained the significant decrease in the incidence of tomato Verticillium wilt in red soil. Moreover, an increase in soil protease activity was also confirmed. Therefore, the application of *Bacillus* could significantly promote the participation of the bacterial community in the phosphorus cycle in red soil, inhibit *V. dahliae*, and promote the growth of tomato plants. Since the soil environment contains biotic and abiotic factors, there are also interactions between them. Therefore, in the future, it will be necessary to further reveal how soil abiotic factors affect the growth and function of *Bacillus* and to identify the key environmental factors affecting the colonization of *Bacillus*. Moreover, it is also necessary to identify the key strains that affect the growth of *Bacillus* in different types of soil environments. The above research can provide further theoretical support for revealing the differences in the application stability of *Bacillus strains*. This requires a wider sampling area and sample, with the help of techniques such as metagenomics and metometabomics.

## 5. Conclusions

We applied *Bacillus* in combination with red soil and found that, compared with black soil, red soil could more effectively inhibit the occurrence of Verticillium wilt and promote phosphorus absorption and growth in tomato plants. In addition, there was a significant difference between the black soil and red soil bacterial communities, which influenced the colonization and function of *Bacillus*. Moreover, the positive regulatory effect of *Bacillus* on the bacterial community in red soil was greater than that in black soil. The application of *Bacillus* significantly promoted the abundance of genes related to antagonistic disease and phosphorus activation in red soil, while the indigenous bacterial community in black soil resisted the effective colonization of exogenous *Bacillus* as well as the positive regulation of structure and function. In the future, we can further use metagenomic and metabolomic analyses to determine the chemical communication between rhizosphere microbial communities and plants after exogenous microbial inoculation to better understand the colonization process and regulatory mechanism of rhizosphere probiotics and to lay a foundation for the use of synthetic microbial agents to manage the plant rhizosphere community, maintain crop health, and increase production.

## Figures and Tables

**Figure 1 microorganisms-12-00797-f001:**
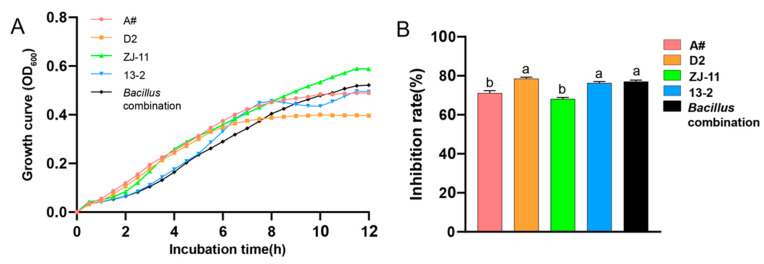
Growth curves of single and combined strains and their ability to inhibit *V. dahliae*. Note: (**A**) growth curves of individual and combined strains; (**B**) inhibition rate of *V. dahliae* by single and combined strains, different letters are used to indicate significant differences between data, such as a and b.

**Figure 2 microorganisms-12-00797-f002:**
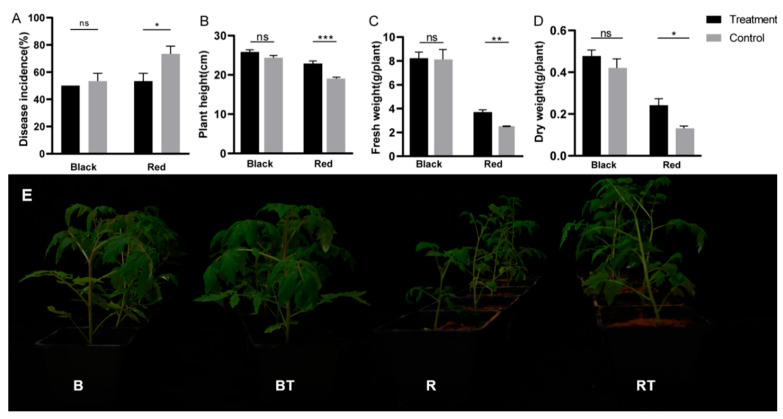
Effects of *Bacillus* on Verticillium wilt incidence in soils with nutrient differences. Note: (**A**) incidence of tomato Verticillium wilt; (**B**) tomato plant height; (**C**) tomato fresh weight; (**D**) tomato dry weight; ns, not significant; * *p* ≤ 0.05, ** *p* ≤ 0.01, *** *p* ≤ 0.001 (*t* test). (**E**) A photo of tomato growth under different treatments. The settings were as follows: *V. dahliae* was added to black soil only (B); *V. dahliae* and *Bacillus* were added to black soil (BT); *V. dahliae* was added to red soil (R); *V. dahliae* and *Bacillus* were added to red soil (RT).

**Figure 3 microorganisms-12-00797-f003:**
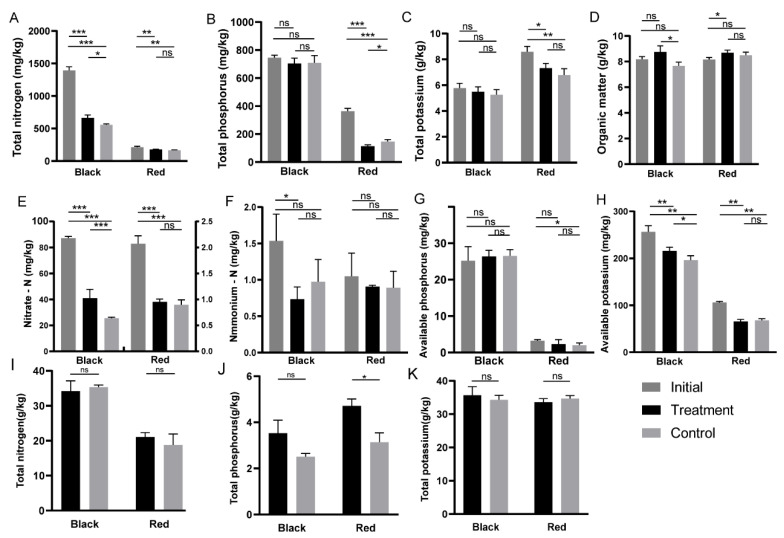
Soil and tomato nutrient contents. Notes: initial, soil sample before planting tomatoes; treatment, soil samples treated with *Bacillus* after planting tomatoes; control, soil samples without *Bacillus* treatment after planting tomatoes. (**A**) Soil total nitrogen content; (**B**) soil total phosphorus content; (**C**) soil total potassium content; (**D**) soil organic matter content; (**E**) soil nitrate nitrogen content; (**F**) soil ammonium nitrogen content; (**G**) soil available phosphorus content; (**H**) soil available potassium content; (**I**) tomato total nitrogen; (**J**) tomato total phosphorus; (**K**) tomato total potassium; ns, not significant; * *p* ≤ 0.05, ** *p* ≤ 0.01, *** *p* ≤ 0.001 (*t* test).

**Figure 4 microorganisms-12-00797-f004:**
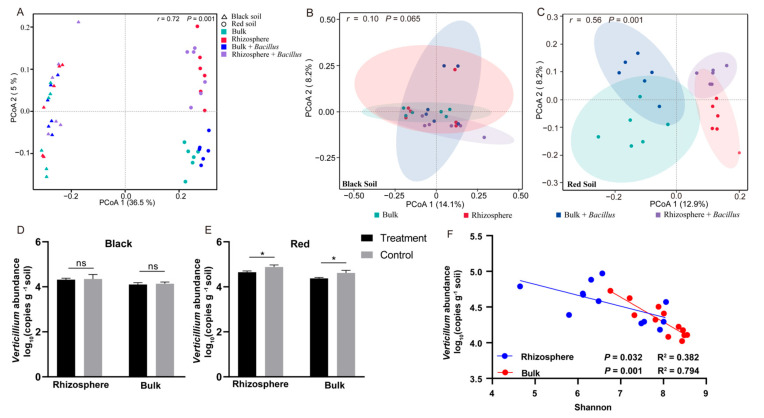
Effect of *Bacillus* application on the bacterial community diversity and abundance of *V. dahliae*. Notes: (**A**) principal coordinate analysis (PCoA) of the bacterial microbial community composition of all the groups; (**B**) principal coordinate analysis (PCoA) of the bacterial community composition among the different groups; (**C**) number of Verticillium species in the soil; (**D**) number of bacteria in black soil; (**E**) effect of soil bacterial diversity on the number of Verticillium species; (**F**) the effect of soil bacterial diversity on the number of *Verticillium*; ns, not significant; * *p* ≤ 0.05 (*t* test).

**Figure 5 microorganisms-12-00797-f005:**
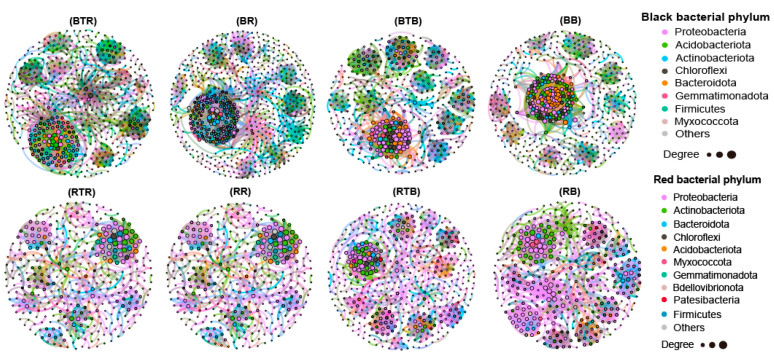
Interaction networks of tomato root soil bacteria under different treatments. Note: BTR, black rhizosphere treatment; BR, black rhizosphere without *Bacillus* inoculation (control); BTB, black bulk treatment; BB, black bulk without *Bacillus* inoculation; RTR, red rhizosphere treatment; RR, red rhizosphere without *Bacillus* inoculation; RTB, red bulk treatment; RB, red bulk without *Bacillus* inoculation.

**Figure 6 microorganisms-12-00797-f006:**
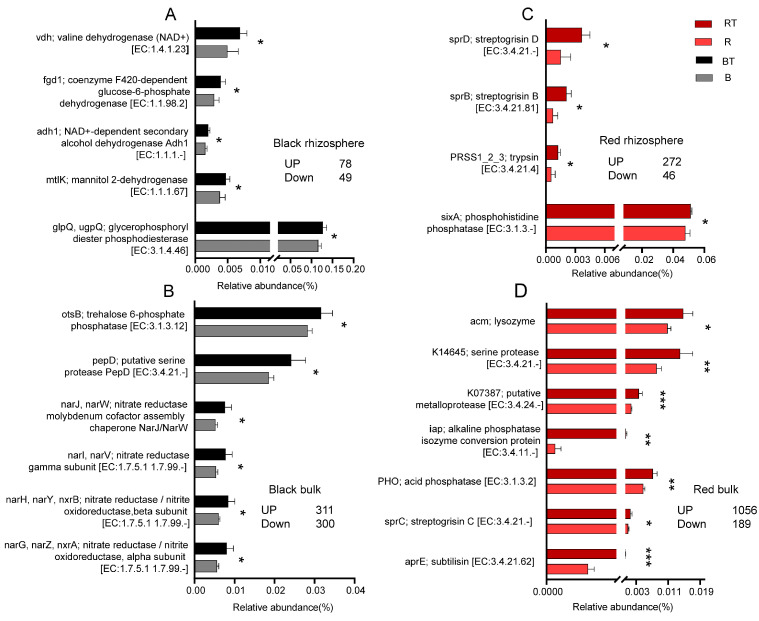
Effects of *Bacillus* application on functional genes. Note: (**A**) black rhizosphere soil; (**B**) black bulk soil; (**C**) red rhizosphere soil; (**D**) red bulk soil; ns, not significant; * *p* ≤ 0.05, ** *p* ≤ 0.01, *** *p* ≤ 0.001 (*t* test).

## Data Availability

Data will be made available on request. The tomato rhizosphere and bulk soil sequencing data obtained in this study are available at the NCBI Sequence Read Archive (https://www.ncbi.nlm.nih.gov/Traces/sra/ (accessed on 8 April 2024)) under accession number PRJNA1061767.

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
