# Peer review of "Difference in the Effect of Applying *Bacillus* to Control Tomato Verticillium Wilt in Black and Red Soil"

_microorganisms, 2024, doi:10.3390/microorganisms12040797_

Round 1

Reviewer 1 Report

Comments and Suggestions for Authors

Review report

The abstract succinctly summarizes the objectives, methods, results, and conclusions of the study. It effectively highlights the key findings regarding the effectiveness of Bacillus in controlling Verticillium wilt and its impact on soil microbial communities.

The introduction provides a clear rationale for the study by addressing the instability of biological control agents in different soil environments. It effectively establishes the importance of understanding Bacillus efficacy in controlling Verticillium wilt in both black and red soils.

The materials and methods section adequately describes the experimental design, including the application of Bacillus in black and red soils and the assessment of Verticillium wilt incidence. However, more details on specific procedures, such as soil sampling techniques and Bacillus application methods, would enhance the reproducibility of the study.

The results section presents the key findings regarding the control effect and intrinsic mechanism of Bacillus in both soil types. It effectively communicates the reduction in Verticillium wilt incidence in red soil and the limited impact in black soil, supported by quantitative data.

The discussion interprets the results within the context of previous literature and theoretical frameworks. It provides insightful explanations for the differential effectiveness of Bacillus in black and red soils, considering factors such as soil microbial communities and nutrient availability.

General Comments:

The study addresses an important issue in agricultural research and provides valuable insights into the effectiveness of Bacillus in controlling Verticillium wilt. The experimental design appears robust, but additional details on certain procedures would improve clarity and reproducibility. The results are well-presented and supported by quantitative data, facilitating a clear understanding of the study findings.

The discussion effectively synthesizes the results and relates them to broader concepts in soil microbiology and plant pathology.

Overall, the manuscript contributes to the scientific understanding of biological control agents in soil environments and warrants consideration for publication pending minor revisions.

Shortcommings in materials and method section of the MS:

Lack of Detailed Soil Characterization: While the study mentions the use of two types of soil with high and low nutritional status, it lacks detailed information about the specific properties of these soils. Important parameters such as pH, organic matter content, and nutrient levels should be provided to allow readers to understand the characteristics of the experimental soils.

Incomplete Description of Microorganism Selection: The section mentions the use of microorganisms screened from tomato roots and those with growth-promoting and disease-preventing functions. However, it does not provide sufficient details about the selection criteria or the specific screening process used to identify these microorganisms. Providing information on the rationale behind the selection of these strains would enhance the transparency and reproducibility of the study.

Limited Information on Growth Curve Determination: The description of growth curve determination lacks clarity regarding the experimental setup and data analysis. Additional details on the specific methodology for measuring bacterial growth, including the duration of the experiment, would improve the comprehensibility of this section.

Unclear Inhibition Effect Analysis: The method for analyzing the inhibition effect of bacterial suspensions on V. dahliae lacks clarity. It would be beneficial to provide more detailed information on the experimental setup, including the criteria used to assess inhibition and the specific measurements taken to determine the radius of mycelium.

Insufficient Information on DNA Extraction and Sequencing: Although the section mentions the extraction of total soil DNA and subsequent sequencing, it lacks detailed information on the specific protocols and techniques used for DNA extraction and sequencing. Providing details on the DNA extraction kit, sequencing platform, and bioinformatics analysis pipeline would improve the reproducibility and transparency of the microbial community analysis.

Limited Explanation of Statistical Analysis: While the section mentions statistical analysis using two-tailed t-tests, it lacks details on the specific comparisons made and the justification for choosing this statistical approach. Providing more information on the rationale behind the statistical methods used and the interpretation of results would enhance the rigor and clarity of the analysis.

Data Availability Statement: The data availability statement mentions the availability of raw sequencing data in the NCBI Sequence Read Archive but does not provide specific accession numbers or links to access the data. Including this information would facilitate reproducibility and data sharing.

Addressing these shortcomings would improve the clarity, reproducibility, and rigor of the Materials and Methods section, thereby enhancing the overall quality of the manuscript.

Reviewer 2 Report

Comments and Suggestions for Authors

The manuscript entitled "Difference in the effect of applying Bacillus to control tomato verticillium wilt in black and red soil" is interesting and has the potential to interest readers. The issues presented in this manuscript are consistent with the topics of the Journal "Microorganisms". The manuscript needs some improvements before publication in the journal "Microorganisms".

Introduction chapter - the current version is quite poor (8 references). Please complete it with the characteristics of Verticillum dahliae and write what losses can be caused by this pathogen in specific environmental conditions. Be specific, not very general as you wrote in lines 62-67. In this chapter you should also state your research hypotheses.

Materials and Methods chapter - the chapter needs to be expanded. In subsection "2.1. Experimental materials' you should give detailed characteristics of the soils used. At least the following parameters must be provided: mechanical composition, organic carbon content, total nitrogen, pH, hydrolytic acidity, sum of exchangeable basic cations, sorption capacity.

Subsections 2.7 and 2.8 should specify the equipment used for the determinations.

Results chapter - the results are correctly described. I would ask you to improve the quality of Figure 6 as it is not very legible now. Also consider showing the types of bacteria, as you did for the phylum in Figure 5.

Discussion and Conclusions chapters - are correctly written and I have no critical comments on them.

References chapter - currently up to 25% of the references are older than the last 10 years. Try to analyse the latest literature more thoroughly. Please make sure that all references cited in the manuscript are included in the reference list and vice versa with the same spelling and dates.

Please check the notation CFU/g throughout the manuscript, e.g. in line 149 you have "3×105 CFU/g", and line 150 "5×106 CFU/g", and I guess it should be 10 to the power of 5 and 10 to the power of 6 respectively.

Reviewer 3 Report

Comments and Suggestions for Authors

Dear authors
Since you write that soil is "an important factor in determining the structure of the microbial community because soil is the habitat of microorganisms and the element content, texture structure, pH, and organic matter content affect the soil microbial community", then how do you interpret the genetic analyses? What are the favourable and unfavourable chemical properties of soils for the growth of Bacillus? Why do you think Bacillus reproduces well in red soil and not in black soil? What I missed in the discussion were the answers to the above questions, in particular which groups of microorganisms (fungi and bacteria) inhibit the development of Bacillus or could inhibit it a priori? What macronutrient levels of N, P, K, Ca and Mg (and perhaps micronutrients) stimulated or inhibited Bacillus development? Did not the resistance of the red soil itself (due to the specific composition of the microorganisms) inhibit Verticilium - the cause of tomato wilt disease? Which pH ranges were favourable (and which were not) for Bacillus growth? So is it possible to influence Bacillus spp. or soil resistance to Verticilium spp. by changing the acidity of the soil?
The below text is a recommendation of what should be included in the discussion (which should not be in the text of the manuscript)? "Authors should discuss the results and how they can be interpreted from the perspective of previous studies and of the working hypotheses. The findings and their implications should be discussed in the broadest context possible. Future research directions may also be highlighted."

Poor legibility of graphs and captions (especially coloured dots).

Comments on the Quality of English Language

The English is generally correct and the manuscript text reads well.
